# Importance of Completing Hybrid Cardiac Rehabilitation for Long-Term Outcomes: A Real-World Evaluation

**DOI:** 10.3390/jcm8030290

**Published:** 2019-02-28

**Authors:** Neville G. Suskin, Salimah Z. Shariff, Amit X. Garg, Jennifer Reid, Karen Unsworth, Peter L. Prior, David Alter

**Affiliations:** 1Lawson Heath Research Institute, Department of Medicine, Division of Cardiology, Western University, London, Ontario, ON N6A 5A5, Canada; 2Institute for Clinical Evaluative Sciences, London, Ontario, ON N6A 5W9, Canada; Salimah.Shariff@ices.on.ca (S.Z.S.); Amit.Garg@lhsc.on.ca (A.X.G.); jennifer.reid@ices.on.ca (J.R.); david.alter@ices.on.ca (D.A.); 3Department of Epidemiology and Biostatistics, Western University, London, Ontario, ON N6A 5C1, Canada; 4Department of Medicine, Division of Nephrology, Western University, London, Ontario, ON N6A 5W9, Canada; 5St. Josephs Health Care, London, Ontario, ON N6A 4V2, Canada; Karen.Unsworth@sjhc.london.on.ca; 6Lawson Heath Research Institute; Department of Psychology, Western University, London, Ontario, ON N6A 4V2, Canada; Peter.Prior@sjhc.london.on.ca; 7University Health Network, Toronto Rehabilitation Institute, Cardiovascular Prevention & Rehabilitation Program, Toronto, Ontario, ON M4G 2V6, Canada; 8Department of Medicine and Institute of Health Policy, Management and Evaluation Health Policy Management & Evaluation, University of Toronto, Toronto, Ontario, ON M5T 3M6, Canada

**Keywords:** cardiac rehabilitation, real-world outcomes

## Abstract

Community-based hybrid cardiac rehabilitation (CR) programs offer a viable alternative to conventional centre-based CR, however their long-term benefits are unknown. We conducted a secondary analysis of the CR Participation Study conducted in London, Ontario, between 2003 and 2006. CR eligible patients hospitalized for a major cardiac event, who resided within 60 min, were referred to a hybrid CR program; 381 of 544 (64%) referred patients initiated CR; an additional 1,498 CR eligible patients were not referred due to distance. For the present study, CR participants were matched using propensity scores to CR eligible non-participants who resided beyond 60 min, yielding 214 matched pairs. Subjects were followed for a mean (standard deviation, SD) of 8.56 (3.38) years for the outcomes of mortality or re-hospitalization for a major cardiac event. Hybrid CR participation was associated with a non-significant 16% lower event rate (Hazard Ratio [HR]: 0.84, 95% CI: 0.59–1.17). When restricting to pairs where CR participants achieved a greater than 0.5 metabolic equivalent exercise capacity increase (123 pairs), CR completion was associated with a 51% lower event rate (HR: 0.49, 95% CI: 0.29–0.81). Successful completion of a community-based hybrid CR program may be associated with decreased long-term mortality or recurrent cardiac events.

## 1. Introduction

Cardiac rehabilitation (CR) is a class 1 or strong recommendation in contemporary clinical practice guidelines following percutaneous coronary intervention (PCI) or coronary artery bypass grafting (CABG) after a myocardial infarction (MI), [1,2,3,4,5,6]. However, participation in CR remains unacceptably low, with the most populous province of Canada (Ontario), providing service to only 18,000 of 53,000 (34%) eligible patients [7]. One reason for this may be the shortage of traditional CR programs with on-site exercise training facilities. To mitigate this gap in access, non-facility-based or community-based CR programming has been recommended [8]. While there is evidence that community-based CR programming that comprises a wide variety of approaches improves risk factor profiles; there is inconsistent randomized controlled trial evidence that these programs reduce mortality or recurrent cardiac events [9,10]. Moreover, the impact of hybrid CR programs where medical assessments occur at a health care institution followed by exercise programming conducted by trained CR professionals at local community exercise facilities is also not well understood [11].

In an effort to increase participation in CR, in 2001 the CR Program in London, Ontario, implemented a hybrid CR model, provided at no-cost to participants. In this model, kinesiologists are integral members of the interdisciplinary CR program and provide individualized exercise training assessment and direct supervision at twice weekly community-centre based exercise training sessions. From 2003 to 2006 a randomized clinical trial (the “CR Participation Study”), was conducted to evaluate the efficacy of usual vs. physician endorsed referral to the London hybrid CR program [12]. The trial collected assessments of all patients experiencing a major cardiac event at two acute-care hospitals in the city, along with details of subsequent CR participation. 

The objective of this study was to perform a secondary analysis of the CR participation study to examine the association between participation in a hybrid CR program on mortality or subsequent major cardiac event [12].

## 2. Methods

### 2.1. Cardiac Rehabilitation (CR) Particpation Study

The CR Participation study was a randomized, single-centre trial that tested the effects of two CR referral processes (usual vs. physician endorsed) to the hybrid CR program situated in London, Ontario [12]. Between May 2003 and September 2006, 2092 patients hospitalized for a myocardial infarction (MI), unstable angina (UA), percutaneous coronary intervention (PCI), or coronary artery bypass grafting (CABG) were screened and deemed eligible for CR (Figure 1); 594 patients were enrolled in the CR Participation study and referred to the hybrid CR program, while 1498 patients were not, as they resided more than a 60-min drive time away from the CR program. Eligibility comprised having experienced an eligible event (MI, UA, PCI or CABG) with no previous CR participation, no planned PCI or CABG within 2 months of hospital discharge, no musculoskeletal problems or stroke preventing exercise training. The hybrid CR program consisted of two weekly on-site exercise training sessions at a community-based exercise facility, in addition to two home-based exercise sessions as guided by the CR kinesiology team over a six-month period. Patients were trained according to the standard FITT (frequency, intensity, time and type) exercise training protocol, consistent with current guidelines [13]. Upon program entry, patients attended an on-site one-on-one assessment by the program cardiologist, along with a CR nurse and kinesiologists, at which time they underwent a peak treadmill exercise test and an exercise prescription was given. In addition, all patients attended at least two sessions with the program dietician to optimize heart healthy nutrition. The program psychology service provided counselling to patients that required or requested psychological intervention for distress or anxiety, as all patients were provided standardised group basic education in psychological factors and cardiovascular conditions, and available CR program psychological services. All patients were asked to complete the Hospital Anxiety and Depression Scale (HADS) for screening purposes [14]. At the hybrid CR intake visit, referral to psychological services was based upon 1) HADS-A or HADS-D > 7 or HADS A+D > 13; or 2) clinician judgement/recommendation; or 3) patient request. Finally, patients’ medications were optimized according to current secondary prevention guidelines [13]. 

### 2.2. Study Design and Data Sources 

The present study is a secondary analysis of the CR Participation study. Patients referred to participate in the hybrid CR program were prospectively entered into the London Cardiovascular Information System (LCVIS), which included clinical and prognostic data. Patients eligible for CR, but not referred were retained in a screening database. These datasets were linked to provincial health administrative databases at the Institute for Clinical Evaluative Sciences (ICES). ICES is an independent, non-profit research institute whose legal status under Ontario’s health information privacy law allows it to collect and analyze health care and demographic data, without consent, for health system evaluation and improvement.

We conducted a retrospective cohort study of patients who initiated the London hybrid CR program matched to non-CR participants between 2003 and 2006. 

We used the LCVIS and screening databases to identify our study subjects and extract information related to exercise capacity (as measured indirectly from treadmill speed and grade by metabolic equivalents or MET score, 1 MET equals 3.5 mL/kg/min oxygen consumption [15]) for CR participants. Subject baseline characteristics, and outcomes were determined using the following health administrative databases: Discharge Abstract Database (DAD), Same Day Surgery (SDS) Database, National Ambulatory Care Reporting System (NACRS), Congestive Heart Failure (CHF), and the Ontario Health Insurance Plan (OHIP). Confirmatory data on cardiovascular procedures including PCI and CABG before and prior to cardiac events were obtained through the OHIP and DAD databases. The use of both DAD and OHIP claims to identify comorbidity and coronary procedures have been previously validated [16,17]. In addition, the presence of diabetes and hypertension prior to cohort inception were identified using disease-specific validated algorithms [18,19]. The Registered Persons Database (RPDB) was used to determine demographic information and mortality. These datasets were linked using unique encoded identifiers and analyzed at ICES. Additional detail on these databases is provided in Appendix A. The International Classification of Diseases 9th and 10th revisions and the Canadian Classification of Health Interventions codes were used to identify all diagnoses and procedures. Variable definitions and administrative codes are detailed in the Appendix A. 

The CR Participation study and the LCVIS database were approved by Western University’s Research Ethics Board (file numbers 08631E and 13045E respectively). Linkage of the screening database to ICES data holdings was approved by Western University’s Research Ethics Board (file number 104999).

### 2.3. Study Sample

We established two cohorts of patients, hybrid CR program participants (CR participants) and non-participants (non-CR participants) using data collected from the CR Participation study and screening database. CR participants were identified as having attended a CR intake appointment between 2003 and 2006. Non-CR participants were included if they were screened and deemed eligible for CR but lived more than a 60-min drive time away from the hybrid CR program and, therefore, were not enrolled in the CR Participation study and hence not referred to the London hybrid CR program. Non-CR participants were excluded if they had more than one screening record. Additionally, patients in both cohorts were excluded if they had no administrative record of hospitalization for the eligible cardiovascular screening event, as additional information about their index hospitalization could not be obtained. Standard data cleaning steps were also applied in both cohorts and full details are shown in Figure 1 and Figure 2. We obtained baseline characteristics (age, sex, socioeconomic status, and index cardiac event) on the cohort entry date (discharge date from initial cardiac event for CR program participants and non-participants); prior cardiac history, comorbidity status, Charlson Comorbidity index [20], and the number of prior hospitalizations were retrieved using administrative records in the five years preceding the cohort entry; and previous encounters with cardiologists and internal medicine specialists in one year prior to the cohort entry. 

### 2.4. Matching Criteria 

We matched non-CR program participants to each CR participant without replacement on age (within 3 years), sex, index cardiac event (MI, PCI, CABG, or unstable angina, UA) and the logit of propensity scores for the predicted probability of CR participation (+/− 0.2 caliper distance of the standard deviation) [21]. Propensity scores were computed using a logistic regression model fitted for the following variables: age, sex, socioeconomic status, index cardiac event; prior cardiac events, co-morbidities (hemorrhagic stroke, stroke or transient ischemic attack (TIA), diabetes, hypertension, atrial fibrillation, peripheral vascular disease, chronic lung disease, chronic kidney disease, cancer, alcoholism, obesity); Charlson co-morbidity; number of prior hospitalizations; number of cardiologist visits, number of internal medicine visits.

For each matched set, we obtained an index date which was the hybrid CR intake date for CR participants and used the same date as the index date for all matched non-CR participants, by assigning the same duration between cohort entry date and hybrid CR program intake date of the matched CR participants. We excluded matched sets where either subject had a subsequent cardiovascular event or procedure (MI, HF, PCI or CABG) (event free), or died prior to the assigned index date. We then randomly selected matched sets without replacement to complete a one to one match. The unmatched subjects were discarded (Figure 2). 

### 2.5. Hybrid CR Program Completion

Hybrid CR Program completion was defined as patients who had a valid exit date, corresponding with an exit clinic, in LCVIS. Successful completion was defined as those individuals who improved their exercise capacity during hybrid CR programming by at least 0.5 metabolic equivalents ((METs), 1 MET = 3.5 mL/kg/min and reflects resting oxygen consumption), the current Canadian quality indicator [22] for CR program exercise improvement. METs were determined indirectly by the peak speed and grade of the treadmill using standard treadmill protocols at hybrid CR program entry and exit peak exercise stress tests [23]. 

### 2.6. Outcomes

The outcome for our primary, secondary and sensitivity analyses was a composite of mortality, or re-hospitalization for MI or HF, or cardiac procedures (PCI or CABG) through 31 December 2015 (Appendix A). Only cardiac procedures that occurred greater than 6-month after index date were considered as procedures that occurred prior to 6 months were assumed to be planned interventions. 

### 2.7. Statistical Analysis

Baseline characteristics were summarized using descriptive statistics; continuous variables as means and standard deviations (SD) and categorical as proportions. Relevant baseline characteristics pre- and post-matching, between the CR participants and non-CR participants were examined using standardized differences. This metric quantifies the differences between group means/proportions relative to the pooled standardized deviation and, when greater than 10%, indicates a potentially meaningful difference [24]. In accordance with ICES privacy policies, cell sizes less than or equal to five were not reported.

A Cox proportional-hazards model, stratified on matched sets, was used to evaluate a composite outcome of death, or re-hospitalization for MI, PCI, CABG, or HF for our primary outcome. Hazard ratios (HR) with 95% confidence intervals (CI) were reported. For our secondary analysis done to account for potential “healthy survivor” bias, we restricted the matched pairs to those where both patients survived at least 1 year (365 days) following the index date and were event free (see definition under matching criteria above) in the 365 days. We performed a sensitivity analysis, to test for potential incremental benefit of attaining a recent Canadian CR quality indicator [22]. For this, we further restricted to pairs that remained event free at the time of CR program exit and where the CR-participant successfully completed CR, defined as a >0.5 MET exercise capacity increase during CR in accordance with the Canadian quality indicator [22].

All statistical analyses were performed using SAS (version 9.3, SAS Institute Inc., Cary, NC, USA). A statistical significance was defined as a two tailed α less than 0.05.

## 3. Results

### 3.1. Baseline Characteristics

After exclusions, 358 hybrid CR program participants and 1192 non-CR participants were eligible for matching (Figure 1). A total of 214 hybrid CR program participants were successfully matched to 214 non-CR program participants (Figure 2). 

The pre-match groups (Appendix A) demonstrated that CR participants differed (standardized difference > 10%) from non-CR participants as they were: younger (mean age 59 vs. 63 years); more likely to be from the highest neighborhood income quintile; less likely to reside in a rural area (7.8% vs. 26%); less likely to have experienced their index cardiac event in the first 2–3 years of patient accrual (2002–2004, 48% vs. 68.8%); less likely to experience UA (5.9% vs. 18.6%) or have undergone CABG (30.7% vs. 37.2%), but more likely to have undergone PCI (45.5% vs. 29.4%) as index cardiac events. As well, CR participants were less likely to have comorbid atrial fibrillation or flutter, hyperlipidemia, chronic kidney disease, peripheral vascular disease, and chronic lung disease; as reflected by a lower prevalence of CR participants in each of the Charlson Comorbidity Index categories. CR participants were also less likely to have been hospitalized within 5 years (30% vs. 80%) or been assessed by a cardiologist (34% vs. 66%) or an internist (59% vs. 91%) at least once within a 1-year period preceding the index cardiac event.

The post-match groups achieved balance and showed no meaningful differences in most measured baseline characteristics apart from place of residence, by design (only 7% CR participants were classified as “rural” residents compared to 28% non-CR participants), neighborhood income quintile (27% of CR participants were in the highest income quintile vs. 22% of non-CR participants), prevalence of peripheral vascular disease (only 2.6% prevalence overall), and year of index event. As well, CR participants remained less likely to have experienced their index cardiac event in the initial 2-3-year period of patient accrual. (Table 1). The mean age of the CR participants and non-CR participants was 61 years, and 26% were females. The majority of patients did not have a prior cardiac background in the 5 years before the index cardiac event, and CABG (40%) was the most common index event, followed by PCI (32%), MI (20%), and UA (8%). 

### 3.2. Outcomes

Subjects had a mean, (SD) of 8.56 (3.38) years of follow-up. (Table 2). CR participation was associated with a non-significant 16% reduction in risk of the primary outcome (HR = 0.84, 95% CI [0.59–1.17]). Adjustment for income quintile and presence of peripheral vascular disease, performed as the standardized difference of these variables between the CR participants and the non-CR participants at baseline remained >10% post matching, did not result in meaningful changes to the analyses (adjusted HR 0.86, 95% CI [0.60–1.22]). The event-free survival probability for the CR participants was non-significantly higher during follow-up compared to non-CR participants (logrank test *p* = 0.080) (Figure 3).

After restricting the matched pairs to those that were event-free at 1-year (secondary analysis), 192 (89.7%) pairs remained. The absolute risk of the primary outcome was non-significantly lower in the CR participants compared to non-CR participants (31.8% vs. 41.2%, HR 0.75, 95% CI [0.52–1.09]) (Table 2). The event-free survival probability for the CR participants was non-significantly higher during follow-up compared to non-CR participants (logrank test *p* = 0.087) (Appendix A).

When restricting to pairs where CR participants successfully completed (≥0.5 MET exercise capacity increase) and were event free at CR exit (sensitivity analysis), 123 (57.5%) pairs remained (Table 2). Accounting for primary outcome events that occurred prior to CR completion, the CR completion attrition rate due to CR non-completion or CR completion without achieving a ≥0.5 MET exercise capacity increase was approximately 40%.

The absolute risk of the primary outcome was lower in the CR participants compared to non-CR participants (26.0% vs. 43.9%), and successful CR completion was associated with a significant 51% lower risk in primary outcome (HR = 0.49, 95% CI [0.29 to 0.81]) (Table 2). The event-free survival probability for the CR participants was significantly greater during follow-up compared to non-CR participants (logrank test *p* = 0.0031) (Figure 4).

## 4. Discussion

This study, to the best of our knowledge, is the first to evaluate real-world outcomes of hybrid CR programming, a CR strategy that can be potentially implemented in any jurisdiction with community-based exercise facilities that have an existing or could establish a relationship with an existing centre-based CR program. Hybrid CR program participation, defined by attendance at the initial CR assessment only, was associated with a non-significant lower risk of dying or re-hospitalization over 10 years for MI, PCI, HF, or CABG. Furthermore, we believe that our study is the first to demonstrate the importance of hybrid CR program completion and achievement of at least a 0.5 MET increase of exercise capacity (Canadian quality indicator) [22] on outcomes, as CR-participants who achieved this experienced a significant 51% lower risk of dying or re-hospitalization over 10 years for MI, PCI, HF, or CABG compared with matched non-CR participants.

Although our investigation was a “real-world” study, the lower risk of dying or re-hospitalization observed for successful CR completers is better interpreted as an “on-treatment efficacy” rather than an “effectiveness” outcome; as it was only noted in approximately 60% of CR participants that successfully completed CR (see “limitations” below). Whether this outcome would have been observed in all CR completers regardless of whether they improved their exercise capacity ≥0.5 MET during CR was not assessed. Nevertheless, out findings are consistent with other large real-world, specialty centre-based CR outcome studies in Calgary and Toronto, where patients who completed CR experienced a 30–50% reduced risk of cardiac rehospitalization or dying compared with patients not referred to CR or patients who were referred to but did not complete CR programming [25,26]. Furthermore, our study measured outcomes over a longer follow-up period than the Calgary study (10 years compared to 5 years) [25].

Our results demonstrate comparable magnitude (as mortality was only one component of our primary outcome), of the benefit of CR programming compared with meta-analyses of randomized clinical trials which have reported a 20–30% mortality reduction associated with specialty centre-based CR compared with usual care [27]. 

Moreover, our study findings are the first to demonstrate the incremental benefits of completing CR programming and achieving the Canadian CR quality indicator exercise capacity increase of >0.5 METs during CR [22]. 

The prospective screening strategy of the CR Participation study and universal referral to CR for CR Participation study subjects, provided reassurance that women were not under-represented in our CR participants as 30% of the pre-match non-CR participants were women compared with 28% of the CR participants (standardized difference 4% or not meaningfully different by convention) and 26% of our post-matched sample were women. Although not the focus of our study, this sets us apart from the Toronto and Calgary studies, where prospective screening was not conducted and women formed only 13% and 17% of their CR participant cohorts, respectively [25,26]. This reinforces the importance of systematic screening for and referral to CR to increase CR participation in women and to the generalizability of our study.

Our study was unique in that it prospectively and consecutively accrued the CR program eligible but non-CR participants (due to their residence being greater than a 60-min drive from the hybrid CR program) and CR program participants as part of the systematic screening strategy for the CR Participation study, which served to decrease selection bias for CR program participation. Moreover, our novel hybrid CR program model utilized existing health care facilities for initial CR participant assessment with subsequent exercise training at a local community-based exercise and wellness facility, supervised by CR program kinesiologists at no cost (including parking) to the CR participants. This hybrid CR model is being used in a number of sites across our province (Ontario) and assuming adherence to CR completion can be enhanced, may be a pragmatic alternative to traditional centre-based or home-based CR programming in a universal payer (government-funded) health care system such as that utilized in Canada [28]. 

Recent original studies or meta-analyses of home or community-based CR programming compared to usual care have not demonstrated consistent mortality or morbidity benefits [11,29]. National CR organizations have surmised that the observed “negative” results may have been due to the study’s outdated CR models, namely deployment of too low an intensity or duration of prescribed exercise training sessions or a lack of accounting for adherence to the prescribed exercise sessions, when measuring outcomes [30]. Assuming patients complete their 6-month CR programming, our hybrid CR program model may be an effective and pragmatic alternative, and may help balance the existing Canadian CR capacity-need mismatch of 1 available CR slot for every 4.5 CR eligible patients [31]. 

Calls for the deployment of better, more flexible models of CR programming, a “rebranding and a reinvigoration” of CR programming [8], to improve patient uptake and completion, have been published lately, including an editorial [32], responding to the recent meta-analysis reporting the positive impact of CR on cardiovascular mortality but not on overall mortality [33]. We and others have reported on the efficacy and need for web-based or virtual CR programming [34,35]. It is conceivable that deployment of regional CR program options, such as specialty centre-based CR, hybrid CR, home-based CR, or virtual CR, that are better aligned with patient choice, may lead to better adherence to whatever CR model patients select.

### Limitations

This study did not include analysis according to a randomized intervention allocation. However, the matching strategy utilized a prospectively collected non-CR participation control that was identified through rigorous screening as part of the CR participation trial. In this way, it is more likely that unmeasured confounders were well-matched between the two groups. Following matching that included a broad socio-economic status and cardiovascular risk-based propensity score, there were no important differences of clinical variables between CR participants and non-CR participants. However, matching resulted in approximately 40% of our CR participants being excluded from our final analyses, which does limit our study’s generalizability.

Our only measure of CR compliance was attendance at the final CR assessment, which prevented us from assessing the impact of the 67% attendance rate of prescribed exercise sessions on outcomes that the Toronto study investigators reported was associated with improved outcomes [26]. We did not specifically assess adherence to prescribed exercise sessions as a priori we selected “CR program completion” based on the “top 5” Canadian quality indicator that addressed CR program completion, namely CR program completion and improved exercise capacity by at least 0.5 METs [22]. However, a post-hoc analysis demonstrated that CR completers who achieved at least a 0.5 MET increase of exercise capacity attended a median (inter-quartile range, IQR) of 32 (21–41) supervised exercise training sessions compared to 25 (5–37) exercise training sessions for CR completers who did not achieve a 0.5 MET increase in exercise capacity (*p* = 0.05). Additionally, CR completers in our study were required to have attended the 6-month exit visit and completed a stress test in order to be included as a “CR completer”.

As the study was a secondary analysis of the CR Participation study [12], patients with musculoskeletal conditions that precluded exercise training, were excluded. This limited our study’s generalizability as studies have shown that musculoskeletal problems are present in 50–60% of hospitalized patients with coronary artery disease who might be eligible for CR [36], however it is unclear how many of these individuals would not have been able to perform exercise training, and hence were excluded in our study.

While an argument that a healthy survivor bias may have accounted for our finding that only successful CR completers experienced a lower event rate than matched non-CR participants, it should be noted that our sensitivity analysis restricted pairs to account for a healthy survivor bias by requiring event-free status at the time of CR completion for CR participants and at a similar time-point for matched non-CR participants. 

Only 60% of CR participants included in the original matched pair analyses remained after restriction to those achieving at least a 0.5 MET increase of exercise capacity and maintenance of event-free status by CR exit. This could further limit the generalizability of our findings, however even if one assumes that this 60% CR completion rate was mainly due to lack of CR completion and not a lack of exercise capacity improvement by at least 0.5 METs (as 5 or less patients experienced a primary outcome event prior to CR exit and were excluded), the 60% CR completion rate is comparable to the recently published CR completion rate of 66% in Ontario [37] and is greater than published rate in Alberta (50%) [25]. 

We did not specifically measure frailty, an important emerging issue for CR patients [38], number of family living together and comorbidity of osteoporotic fracture, factors that may have negatively impacted our hybrid CR program completion rate. However, our matching procedure included a number of variables that contribute to these factors such as age, socioeconomic status and the Charlson comorbidity index. Consequently, we anticipate that the frequency of these factors was partially balanced between our CR participants and non-participants.

## 5. Conclusions

Community-based hybrid CR program completion may be associated with decreased long-term mortality or recurrent cardiac events. Future study is needed to improve adherence to and to assess potential health care cost benefits of the hybrid-CR program delivery model from the perspective of a universal health care provider system such as that available in Canada. 

## Figures and Tables

**Figure 1 jcm-08-00290-f001:**
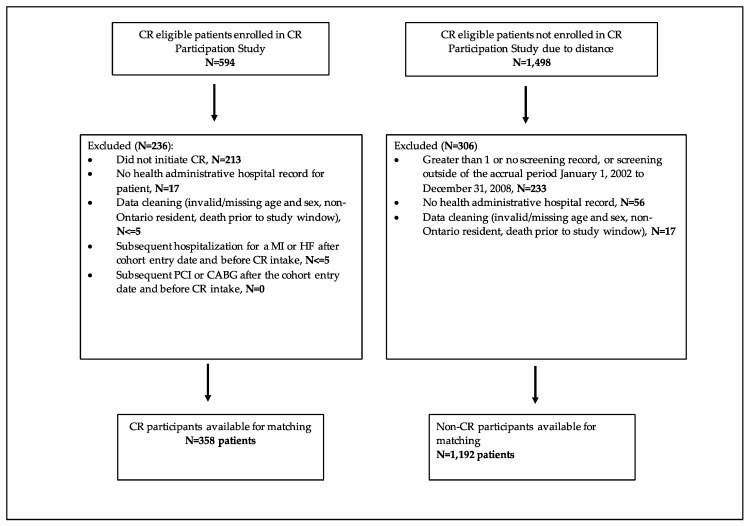
Study flow—selection. CR = cardiac rehabilitation, MI = myocardial infarction, HF = heart failure, PCI = percutaneous coronary intervention, CABG = coronary artery bypass grafting.

**Figure 2 jcm-08-00290-f002:**
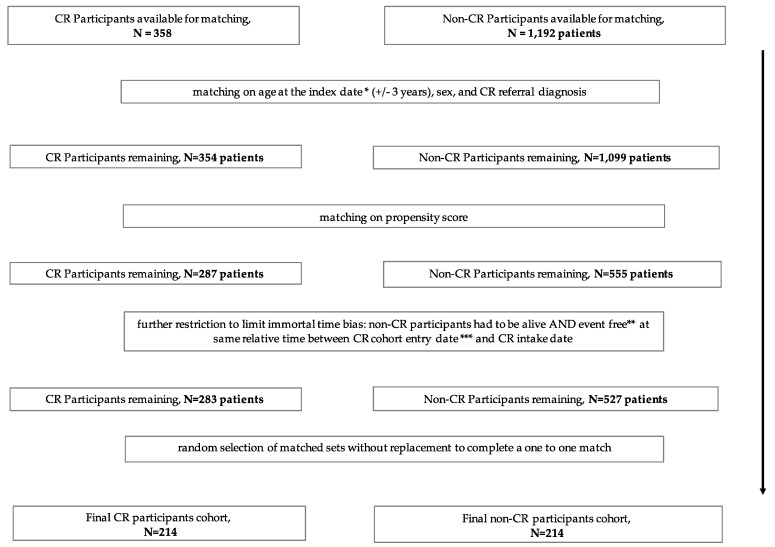
Study flow—matching. * The hybrid CR intake date for CR participants and matched “fake” intake date for all matched non-CR participants, ** patients had no subsequent cardiovascular event or procedure (MI, HF, PCI, CABG), *** discharge date from initial cardiac event for CR program participants and non-participants, CR = cardiac rehabilitation, MI = myocardial infarction, HF = heart failure, PCI = percutaneous coronary intervention, CABG = coronary artery bypass grafting.

**Figure 3 jcm-08-00290-f003:**
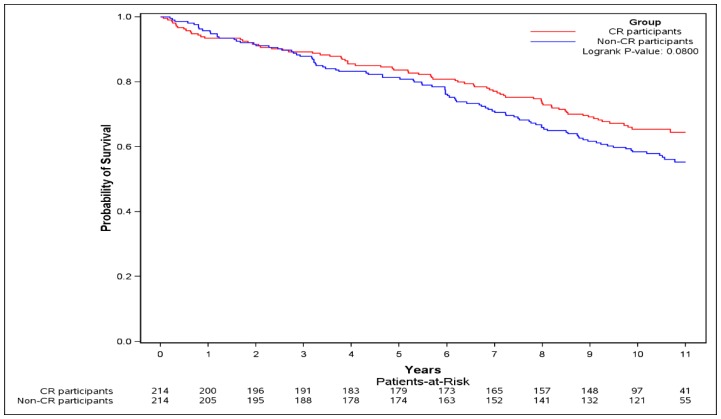
Kaplan–Meier curve of the primary outcome. The Kaplan–Meier curve of the primary outcome (composite of death, or re-hospitalization for myocardial infarction (MI), or percutaneous coronary intervention (PCI), or coronary artery bypass grafting (CABG), or heart failure (HF)) was plotted and logrank test was performed.

**Figure 4 jcm-08-00290-f004:**
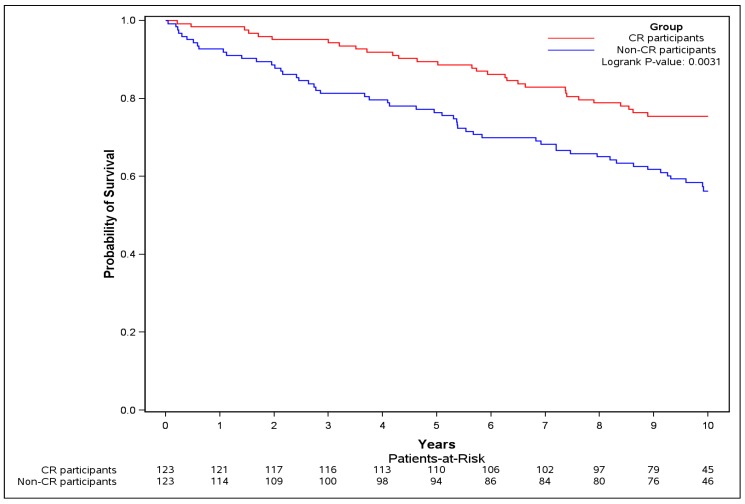
Kaplan–Meier curve of overall survival for sensitivity analysis. The Kaplan–Meier curve for the primary outcome (composite of death, or re-hospitalization for MI, or PCI, or CABG, or HF) was plotted and logrank test was performed, restricting to pairs where CR participants successfully completed (≥0.5 MET exercise capacity increase) and were event free at CR exit.

**Table 1 jcm-08-00290-t001:** Post-match baseline patient characteristics.

	Non-CR Participants (*n* = 214)	CR Participants (*n* = 214)	Total (*n* = 428)	Standardized Difference ^1^
Demographics				
Age (years), mean (SD)	60.89 ± 9.15	60.86 ± 9.07	60.87 ± 9.10	0
Female sex, *n* (%)	56 (26.2%)	56 (26.2%)	112 (26.2%)	0
Income quintile, *n* (%)				
Quintile 1	30 (14.0%)	32 (15.0%)	62 (14.5%)	0.03
Quintile 2	55 (25.7%)	45 (21.0%)	100 (23.4%)	0.11
Quintile 3	36 (16.8%)	41 (19.2%)	77 (18.0%)	0.06
Quintile 4	47 (22.0%)	39 (18.2%)	86 (20.1%)	0.09
Quintile 5	46 (21.5%)	57 (26.6%)	103 (24.1%)	0.12
Rural, yes, *n* (%)	59 (27.6%)	15 (7.0%)	74 (17.3%)	0.56
Year of cohort entry, *n* (%)				
2003	25 (11.7%)	15 (7.0%)	40 (9.3%)	0.16
2004	109 (50.9%)	91 (42.5%)	200 (46.7%)	0.17
2005	≤80	≤70	144 (33.6%)	0.14
2006	≤5	≤45	44 (10.3%)	0.68
Index cardiac event, *n* (%)				
Myocardial infarction	43 (20.1%)	43 (20.1%)	86 (20.1%)	0
Unstable angina	16 (7.5%)	16 (7.5%)	32 (7.5%)	0
Percutaneous coronary intervention	69 (32.2%)	69 (32.2%)	138 (32.2%)	0
Coronary artery bypass graft surgery	86 (40.2%)	86 (40.2%)	172 (40.2%)	0
Time between hospital discharge and start of CR (days)				
Mean (SD)	103.28 (47.67)	103.28 (47.67)	103.28 (47.62)	0
Prior cardiac events in the previous 5 years, *n* (%)				
Myocardial infarction	26 (12.1%)	22 (10.3%)	48 (11.2%)	0.06
Unstable angina	22 (10.3%)	21 (9.8%)	43 (10.0%)	0.02
Percutaneous coronary intervention	7 (3.3%)	11 (5.1%)	18 (4.2%)	0.09
Coronary artery bypass graft surgery	0	0	0	
Heart failure	9 (4.2%)	9 (4.2%)	18 (4.2%)	0
Comorbidities in the previous 5 years, *n* (%)				
Atrial fibrillation/flutter	≤5	≤5	≤10	0
Hypertension	51 (23.8%)	53 (24.8%)	104 (24.3%)	0.02
Hyperlipidemia	9 (4.2%)	6 (2.8%)	15 (3.5%)	0.08
Haemorrhagic stroke	0 (0.0%)	≤5	≤5	0.1
Ischemic stroke	0 (0.0%)	≤5	≤5	0.1
Transient ischemic stroke	≤5	≤5	≤5	0
Chronic kidney disease	11 (5.1%)	13 (6.1%)	24 (5.6%)	0.04
Diabetes mellitus	24 (11.2%)	18 (8.4%)	42 (9.8%)	0.09
Peripheral vascular disease	≤10	≤5	11 (2.6%)	0.15
Chronic lung disease (including chronic obstructive pulmonary disease)	45 (21.0%)	44 (20.6%)	89 (20.8%)	0.01
Major cancers	16 (7.5%)	14 (6.5%)	30 (7.0%)	0.04
Alcoholism	≤5	≤5	≤5	0.06
Obesity	≤5	≤5	≤10	0
Charlson comorbidity index [20]				
0,1	63 (29.4%)	63 (29.4%)	126 (29.4%)	0
2	10 (4.7%)	12 (5.6%)	22 (5.1%)	0.04
3+	14 (6.5%)	9 (4.2%)	23 (5.4%)	0.1
No hospitalizations	127 (59.3%)	130 (60.7%)	257 (60.0%)	0.03
Healthcare system utilization, *n* (%)				
Hospital episodes				
0	127 (59.3%)	130 (60.7%)	257 (60.0%)	0.03
1–5	84 (39.3%)	82 (38.3%)	166 (38.8%)	0.02
6+	≤5	≤5	≤5	0.04
Visits to a cardiologist				
0	100 (46.7%)	103 (48.1%)	203 (47.4%)	0.03
1+	114 (53.3%)	111 (51.9%)	225 (52.6%)	0.03
Visits to an internist				
0	57 (26.6%)	49 (22.9%)	106 (24.8%)	0.09
1+	157 (73.4%)	165 (77.1%)	322 (75.2%)	0.09

^1^, Standardized difference where meaningful difference is greater than 0.1; Abbreviations: CR, hybrid cardiac rehabilitation; SD, standard deviation.

**Table 2 jcm-08-00290-t002:** Outcomes.

	Non-CR Participants	CR Participants	*p*-Value ^1^
Primary analysis ^2^			
Number of subjects	214	214	-
Follow-up duration in years Median (IQR)	10.38 (6.17–11.02)	9.88 (7.81–10.84)	-
Events *N* (%)	97 (45.3)	76 (35.5)	-
HR ^3^ unadjusted (95% CI)	-	0.84 (0.60–1.17)	0.3
HR ^4^ adjusted (95% CI)	-	0.86 (0.60–1.22)	0.39
Secondary analysis (event-free at 1-year) ^5^			
Number of subjects (%)	192 (matched)	192 (89.7%)	-
Follow-up duration in years Median (IQR)	9.48 (6.20–10.05)	8.92 (7.49–9.85)	
Events N (%)	79 (41.1%)	61 (31.8)	-
HR ^4^ unadjusted (95% CI)	-	0.75 (0.52–1.09)	0.13
Sensitivity analysis (event-free at successful CR completion) ^6^			
Number of subjects (%)	123 (matched)	123 (57.5%)	-
Follow-up duration in years Median (IQR)	9.72 (5.30–10.37)	9.51 (8.54–10.32)	
Events *N* (%)	54 (43.9)	32 (26.0)	-
HR ^4^ unadjusted (95% CI)	-	0.49 (0.29–0.81)	0.006

^1^, 2-sided p with significance set at <0.05; ^2^, The primary analysis assessed for the composite outcome of death, or re-hospitalization for MI, or PCI, or CABG, or HF during follow-up; ^3^, The hazard ratio of the outcome comparing CR participants vs. non-CR participants was derived from the Cox proportional hazards model without including any baseline covariates. The proportional hazard assumption was assessed before model fitting; ^4^, The Hazard ratio of the outcome comparing CR participants vs. non-CR participants was derived from Cox proportional hazards model including income quintile and peripheral vascular disease covariates. The proportional hazard assumption was assessed before model fitting. As the adjusted hazard ratio was similar to the unadjusted hazard ratio, subsequent hazard ratios (secondary and sensitivity analyses below) were not adjusted; ^5^, The secondary analysis assessed for the composite outcome of death, or re-hospitalization for MI, or PCI, or CABG, or HF during follow-up restricting the sample to those pairs who were event-free at 1-year after index date (CR entry for CR participant or matched date for non-CR participant); ^6^, The sensitivity analysis assessed for the composite outcome of death, or re-hospitalization for MI, or PCI, or CABG, or HF during follow-up restricting the sample to pairs that remained event free at the time of CR program exit and where the CR-participant successfully completed CR, defined as a >0.5 MET exercise capacity increase during CR in accordance with the Canadian quality indicator [22]. Abbreviations: IQR, inter-quartile range.

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
