# Peer review of "Importance of Completing Hybrid Cardiac Rehabilitation for Long-Term Outcomes: A Real-World Evaluation"

_jcm, 2019, doi:10.3390/jcm8030290_

Round 1
Reviewer 1 Report
Suskin and co-workers demonstrated the effectiveness of rehabilitation program by multidisciplinary, and the participants accomplished improved survival, compared with non-program participants. It is interesting, but this reviewer wonders the outcome may be influenced by multiple factors. Specifically, it is interested to include the status of frailty, number of family living together, and comorbidity of osteoporotic fracture. They also significantly contribute to adherence to accomplish the program.
Author Response
We wish to thank the editors for requesting minor reviews for our paper based on the thoughtful comments of the 2 reviewers. Responses are categorized by reviewer and structured as “Reviewer’s comments” and “Response” where appropriate.
Thank you for indicating that the: introduction provided sufficient background and included all relevant references, the research design was appropriate, the methods were adequately described, and that the conclusions were supported by the results.
Reviewer’s comments:
· It is interesting, but this reviewer wonders the outcome may be influenced by multiple factors. Specifically, it is interested to include the status of frailty, number of family living together, and comorbidity of osteoporotic fracture. They also significantly contribute to adherence to accomplish the program.
Response:
· We concur that these factors may have influenced our adherence rates to the hybrid CR program, unfortunately due to the administrative-nature of our data, these variables are unavailable to us. We have now included the following sentences beginning on line 366 in our limitation section to address this issue: “We did not specifically measure frailty, an important emerging issue for CR patients (Vigorito et al. 2017), number of family living together and comorbidity of osteoporotic fracture, factors which may have negatively impacted our hybrid CR program completion rate. However, our matching procedure included a number of variables that contribute to these factors such as age, socioeconomic status and the Charlson comorbidity index. Consequently, we anticipate that the frequency of these factors was partially balanced between our CR participants and non-participants.”
We trust that this response is satisfactory.
References
Vigorito, C., A. Abreu, M. Ambrosetti, R. Belardinelli, U. Corra, M. Cupples, C. H. Davos, S. Hoefer, M. C. Iliou, J. P. Schmid, H. Voeller & P. Doherty (2017) Frailty and cardiac rehabilitation: A call to action from the EAPC Cardiac Rehabilitation Section. European Journal of Preventive Cardiology, 24, 577-590
Reviewer 2 Report
This is a well-written paper on 'hybrid CR' and long term outcomes. There are a small number of points that the authors may wish to address, however.
It is somewhat unusual to define 'completion' in terms of an outcome achieved, without explicit reference to number of sessions completed. Could the authors say something about the relationship between these two?
I am concerned at the exclusion of people with musculoskeletal conditions in a real-world. Marzolini S, et al. Musculoskeletal comorbidities in cardiac patients: prevalence, predictors, and health services utilization. Arch Phys Med Rehabil 2012; 93: 856-862. showed in a sample of 1,800 Canadian CHD patients, a higher rate of musculoskeletal (MSK) conditions than many other studies eg in the UK and Australia, (over 50%), but demonstrated that MSK conditions were associated with depression, obesity, lower socioeconomic status (SES) and higher health services utilisation.
In Australia, 2017-2018 Health Survey data show 57.5% of CVD patients with arthritis and 34% with back problems. Could the authors perhaps consider this exclusion as a limitation in their study?
Is the structure of the CR typical in this jurisdiction, with apparently no standardised educational component for ALL participants on psychosocial factors, but rather a response to requests for counselling?
Good to see the attention paid to recruitment of women.
Author Response
We wish to thank the editors for requesting minor reviews for our paper based on the thoughtful comments of the 2 reviewers. Responses are categorized by reviewer and structured as “Reviewer’s comments” and “Response” where appropriate.
Thank you for indicating that the: introduction provided sufficient background and included all relevant references, research design was appropriate, methods were adequately described, results were clearly presented, and the conclusions were supported by the results.
Reviewer’s comment:
· This is a well-written paper on 'hybrid CR' and long-term outcomes.
Response:
· Thank you.
Reviewer’s comment:
· It is somewhat unusual to define 'completion' in terms of an outcome achieved, without explicit reference to number of sessions completed. Could the authors say something about the relationship between these two?
Response:
· Thank-you for this comment. We have added the following paragraph starting on line 337. “We did not specifically assess adherence to prescribed exercise sessions as we a priori selected “CR program completion” based on the “top 5” Canadian quality indicator that addressed CR program completion, namely CR program completion and improved exercise capacity by at least 0.5 METs (Grace et al. 2014). However, a post-hoc analysis demonstrated that CR completers who achieved at least a 0.5 MET increase of exercise capacity attended a median (IQR) of 32 (21-41) supervised exercise training sessions compared to 25 (5-37) exercise training sessions for CR completers who did not achieve a 0.5 MET increase in exercise capacity (p=0.05). Additionally, CR completers in our study were required to have attended the 6-month exit visit and completed a stress test in order to be included as a “CR completer”.
Reviewer’s comment:
· I am concerned at the exclusion of people with musculoskeletal conditions in a real-world. Marzolini S, et al. Musculoskeletal comorbidities in cardiac patients: prevalence, predictors, and health services utilization. Arch Phys Med Rehabil 2012; 93: 856-862. showed in a sample of 1,800 Canadian CHD patients, a higher rate of musculoskeletal (MSK) conditions than many other studies eg in the UK and Australia, (over 50%), but demonstrated that MSK conditions were associated with depression, obesity, lower socioeconomic status (SES) and higher health services utilisation. In Australia, 2017-2018 Health Survey data show 57.5% of CVD patients with arthritis and 34% with back problems. Could the authors perhaps consider this exclusion as a limitation in their study?
Response:
· We concur, and now have included this as a limitation in line 347, as follows. “As this study was a secondary analysis of the CR Participation study (Suskin et al. 2007) patients with musculoskeletal conditions that precluded exercise training, were excluded. This limits our study’s generalizability as studies have shown that musculoskeletal problems are present in 50-60% of hospitalized patients with coronary artery disease who might be eligible for CR (Marzolini et al. 2012), however it is unclear how many of these individuals would not have been able to perform exercise training, and hence were excluded in our study.”
Reviewer’s comments:
· Is the structure of the CR typical in this jurisdiction, with apparently no standardised educational component for ALL participants on psychosocial factors, but rather a response to requests for counselling?
Response:
· Thank you for bringing this to our attention. To the best of our knowledge the vast majority (>95%) of CR programs in our province of Ontario and in Canada, provide stress management or psychological services and patient education (Tran et al. 2018). We acknowledge that our description of our hybrid CR program’s psychological assessment and services was incomplete - an updated description has been inserted in line 79 as follows. “..., as all patients were provided standardised group basic education in psychological factors and cardiovascular conditions, and available CR program psychological services. All patients were asked to complete the Hospital Anxiety & Depression Scale (HADS) for screening purposes (Zigmond and Snaith 1983). At the hybrid CR intake visit, referral to psychological services was based upon 1) HADS-A or HADS-D > 7 or HADS A+D >13; or 2) clinician judgement / recommendation; or 3) patient request”
Reviewer’s comments:
· Good to see the attention paid to recruitment of women.
Response:
Thank you
References
Grace, S. L., P. Poirier, C. M. Norris, G. H. Oakes, D. S. Somanader, N. Suskin & C. A. o. C. Rehabilitation (2014) Pan-Canadian development of cardiac rehabilitation and secondary prevention quality indicators. Can J Cardiol, 30, 945-8.
Marzolini, S., P. I. Oh, D. Alter, D. E. Stewart, S. L. Grace & C. R. C. C. t. A. R. E. Investigators (2012) Musculoskeletal comorbidities in cardiac patients: prevalence, predictors, and health services utilization. Arch Phys Med Rehabil, 93, 856-62.
Suskin, N., J. Irvine, J. M. O. Arnold, R. Turner, J. Zandri, P. Prior & K. Unsworth (2007) Improving cardiac rehabilitation (CR) participation in women and men, the CR participation study. Journal of cardiopulmonary rehabilitation, 27, 342-342.
Tran, M., E. Pesah, K. Turk-Adawi, M. Supervia, F. L. Jimenez, P. Oh, C. Baer & S. L. Grace (2018) Cardiac Rehabilitation Availability and Delivery in Canada: How Does It Compare With Other High-Income Countries? Canadian Journal of Cardiology, 34, S252-S262.
Zigmond, A. S. & R. P. Snaith (1983) The hospital anxiety and depression scale. Acta Psychiatrica Scandinavica, 67, 361-370.